# Speech Processing for Language Learning: A Practical Approach to Computer-Assisted Pronunciation Teaching



Natalia Bogach [1,*,†], Elena Boitsova [1,†], Sergey Chernonog [1,†], Anton Lamtev [1,†], Maria Lesnichaya [1,†], Iurii Lezhenin [1,†], Andrey Novopashenny [1,†], Roman Svechnikov [1,†], Daria Tsikach [1,†], Konstantin Vasiliev [1,†], Evgeny Pyshkin [2,*,†] and John Blake [3,*,†]

1    Institute of Computer Science and Technology, Peter the Great St. Petersburg Polytechnic University, 195029 St. Petersburg, Russia; el-boitsova@yandex.ru (E.B.); chernonog.sa@edu.spbstu.ru (S.C.); lamtev.au@edu.spbstu.ru (A.L.); lesnichaya.md@edu.spbstu.ru (M.L.); lezhenin.ii@edu.spbstu.ru (I.L.); novopashenny.ag@edu.spbstu.ru (A.N.); svechnikov.ra@edu.spbstu.ru (R.S.); tsikach.da@edu.spbstu.ru (D.T.); vasilievKonst@mail.ru (K.V.)
2    Division of Information Systems, School of Computer Science and Engineering, University of Aizu, Aizuwakamatsu 965-8580, Japan
3    Center for Language Research, School of Computer Science and Engineering, University of Aizu, Aizuwakamatsu 965-8580, Japan
*    Correspondence: bogach@kspt.icc.spbstu.ru (N.B.); pyshe@u-aizu.ac.jp (E.P.); jblake@u-aizu.ac.jp (J.B.)
†    All authors contributed equally to this work.

**Abstract:** This article contributes to the discourse on how contemporary computer and information technology may help in improving foreign language learning not only by supporting better and more flexible workflow and digitizing study materials but also through creating completely new use cases made possible by technological improvements in signal processing algorithms. We discuss an approach and propose a holistic solution to teaching the phonological phenomena which are crucial for correct pronunciation, such as the phonemes; the energy and duration of syllables and pauses, which construct the phrasal rhythm; and the tone movement within an utterance, i.e., the phrasal intonation. The working prototype of StudyIntonation Computer-Assisted Pronunciation Training (CAPT) system is a tool for mobile devices, which offers a set of tasks based on a "listen and repeat" approach and gives the audio-visual feedback in real time. The present work summarizes the efforts taken to enrich the current version of this CAPT tool with two new functions: the phonetic transcription and rhythmic patterns of model and learner speech. Both are designed on a base of a third-party automatic speech recognition (ASR) library Kaldi, which was incorporated inside StudyIntonation signal processing software core. We also examine the scope of automatic speech recognition applicability within the CAPT system workflow and evaluate the Levenstein distance between the transcription made by human experts and that obtained automatically in our code. We developed an algorithm of rhythm reconstruction using acoustic and language ASR models. It is also shown that even having sufficiently correct production of phonemes, the learners do not produce a correct phrasal rhythm and intonation, and therefore, the joint training of sounds, rhythm and intonation within a single learning environment is beneficial. To mitigate the recording imperfections voice activity detection (VAD) is applied to all the speech records processed. The try-outs showed that StudyIntonation can create transcriptions and process rhythmic patterns, but some specific problems with connected speech transcription were detected. The learners feedback in the sense of pronunciation assessment was also updated and a conventional mechanism based on dynamic time warping (DTW) was combined with cross-recurrence quantification analysis (CRQA) approach, which resulted in a better discriminating ability. The CRQA metrics combined with those of DTW were shown to add to the accuracy of learner performance estimation. The major implications for computer-assisted English pronunciation teaching are discussed.

**Keywords:** speech processing; computer-assisted pronunciation training (CAPT); voice activity detection (VAD); audio-visual feedback; time warping (DTW); cross-recurrence quantification analysis (CRQA)

## 1. Introduction

In the last few decades, digital technology has transformed educational practices. This process continues to evolve with technology-driven digitalization fostering a completely new philosophy for both teaching and learning. Advancements in computer technology and human-computer interaction (HCI) applications have enabled digitally driven educational resources to go far beyond the mere digitization of the learning process. CAPT systems belong to the relatively new domain of technology and education, which is why there are still many open issues. However, from the perspective of our current understanding we hold that focusing on computer-assisted pronunciation teaching is beneficial to language education as it helps to involve the very basic cognitive mechanisms of language acquisition. In this work we describe our efforts to pursue this process because

> ... pronunciation is not simply a fascinating object of inquiry, but one that permeates all spheres of human life, lying at the core of oral language expression and embodying the way in which the speaker and hearer work together to produce and understand each other... [1].

Language learning has greatly benefitted from global technological advancement. With the appearance of mobile personalized devices, the extensive use of visual perception in conjunction to the audial input has become easier to implement. Online apps enable learners to use their mobile phones to improve their language skills in and outside classes. Apps that are not mobile-friendly are likely to be shunned by learners, particularly those whose primary porthole to the Internet is through their mobile device. From a technological perspective, mobile-first approach and user-friendly interface favoring networking in both technical and communication senses facilitate the building of highly accessible apps aimed at improving pronunciation anytime and anywhere [2].

Native speaker pronunciation models have dominated English language teaching for many years. However, there is now increasing acceptance of world Englishes and non-standard pronunciation (i.e., not adhering to Anglophonic norms), stemming from the three concentric circle model proposed by Kachru [3]. His model places historical and sociolinguistic bases of varieties of English, such as American and British English, in the inner circle surrounded by the outer circle of countries that use English as a *lingua franca*, such as India and Kenya, while the expanding circle comprises countries that use English as medium to communicate internationally. With fewer users of English in the inner circle than the other two circles, there is debate over the appropriacy of native-speaker models. The trend to focus on intelligibility originated in the 1990s. Those arguing against native-speaker models note that intelligibility should be paramount. In recent years, the prevailing approach holds that intelligibility is more important than replicating native-speaker models [4–7]. Rightly or wrongly the goal of many learners of English is to emulate native-speaker models of pronunciation. This may be due to its perceived status and pragmatic value [8].

Intonation patterns that do not conform to Anglocentric expectations (i.e., native speaker norms) are more likely to affect a native speaking hearer's judgement of the attitude of the speaker rather than causing misunderstanding over the content of the message. For example, inappropriate pitch trajectories in English, such as using low pitch displacement for requests, are likely to be interpreted as rude by native speakers [9]. Rhythm is used by native speakers to focus attention on certain key words and so when non-Anglocentric patterns are used, the comprehensibility of the message drops, forcing empathetic native speakers to concentrate intensively to decode the message. The intonation contour also

indicates the pragmatic status of contextualized information as either given (known) or new (unknown) [10–12]. Specifically, Halliday noted that new information is marked by tonic pitch movement [13,14]. This enables speakers to convey new information with more clarity [15], reducing the burden on listeners to work out the words to which more attention should be paid. Communication competence in a foreign language and the ability to understand foreign language in its natural setting is much affected by the speaker's level of pronunciation [16].

Most language teachers focus learners on a particular aspect of pronunciation and then provide opportunities for them to practise [17]. This approach follows the noticing hypothesis [18], which states that noticing is a necessary precursor to learning. The effects of different attentional states and contexts with their pedagogical implications have been further examined in [19] where the relation between the attention state and the stimuli were shown to vary with context. The complexity of phonological problems combined with simultaneously teaching learners with different phonological problems exacerbates the difficulty for classroom teachers to provide personalised feedback, particularly for those who teach large classes [20].

The ideal solution is one that could present feedback to learners in their preferred form, e.g., graphical, textual or audio feedback. Graphical feedback requires learners to identify the key differences themselves. A feature that provides specific actionable advice in text form on differences between target language and learner performance by comparing the pitch waves could enhance the usability. Some learners prefer to hear than read, and so providing that option would be well received by auditory learners.

The CAPT system in focus is StudyIntonation [21–23]. The practical purpose of the StudyIntonation project is twofold: first, to develop and assess a technology-driven language learning environment including a course toolkit with end-user mobile and web-based applications; and second, to develop tools for speech annotation and semantic analysis based on intonation patterns and digital signal processing algorithms. There has been significant advances in segmental pronunciation training. A framework proposed in [24] detects mispronunciations and maps each case to a a phone-dependent decision tree where it can be interpreted and communicated to the second language learners in terms of the path from a leaf node to the root node. Similar systems, which address teaching segmental and prosodic speech phenomena, have been developed for different languages and groups of learners. Systems which are most similar to StudyIntonation in terms of architecture and functionality were developed for suprasegmental training in English [25], for children with profound hearing loss to learn Hungarian [26], and for Italian adults learning English as a foreign language [27]. This study, in particular, exemplifies and explains the technological impact on CAPT development. These and many more CAPT systems adopt a common approach, which is based on speech processing, segmentation and forced alignment to obtain some numerically and graphically interpretable data for a set of pronunciation features [28]. The individual systems or solutions differ in their choice of pronunciation aspects to train and scoring policy: a conventional approach is the simultaneous visual display of model and learner's rhythm and intonation and the scoring performed as per Pearson and/or Spearman correlation or dynamic time warping (DTW). Even very powerful CAPT tools are still lacking explicit feedback for acquisition and assessment of foreign language suprasegmentals [29].

Intonation contour and rhythmic portrait of a phrase provide learners with a better understanding of how they follow the recorded patterns of native speakers. However, such graphs do not presume innate corrective or instructive value. Conventional score-based approach cannot tell the second language learners why their mispronunciations occur and how to correct them. Consequently, adequate metrics to estimate learning progress and prosody production should be combined with CAPT development, while giving more intuitive and instructive feedback. Time-frequency and cepstrum prosody features, which are fairly well suited to the purpose of automatic classification, are impractical to grasp synchronization and coupling effects during learner interaction with a CAPT system.

That is why the approach based on non-linear dynamics theory, in particular, recurrence quantification analysis (RQA) and cross RQA (CRQA) may contribute to the adequacy of provided feedback.

In this work we applied voice activity detection (VAD) before pitch processing and instrumented StudyIntonation with a third-party automatic speech recognition (ASR) system. ASR internal data obtained at intermediary stages of speech to text conversion provide phonetic transcriptions of the input utterances of both the model and learner. The rhythmic pattern is retrieved from phonemes and their duration and energy. Transcription and phrasal rhythm are visualized alongside with phrasal intonation shown by pitch curves. CAPT courseware is reorganized to represent each task as a hierarchical phonological structure which contains an intonation curve, a rhythmic pattern (based on energy and duration of syllables) and IPA transcription. The validity of dynamic time-warping (DTW), which is currently applied to determine the prosodic similarity between models and learners was estimated over native and non-native speakers corpora using IViE stimuli. DTW results were compared and contrasted against CRQA metrics which measured synchronization and coupling parameters in the course of CAPT operation between model and learner; and were, thus, shown to add to the accuracy of learner performance evaluation through the experiments with two automatic binary classifiers.

Our research questions are aimed at clarification of the (1) ASR system and ASR speech model applicability within a mobile CAPT system; (2) ASR system internal data consistency to represent phonological events for teaching purposes; and (3) CRQA impact to learner performance evaluation.

The remainder of this paper is structured as follows. Section 2 is a review of the literature relevant to the cross-disciplinary research of CAPT system design, Section 3 provides an overview of speech processing methods which extend the existing prototype functionality (voice activity detection; phoneme, syllable, and rhythm retrieval algorithms). Our approach to conjoin different phonological aspects of pronunciation within the learning content of the CAPT system is also described. Section 4 contains the CAPT system signal processing software core design and presents the stages of its experimental assessment with reference phonological corpora and during target user group try-outs. Section 5 discusses the major implications of our research. Specifically, the accuracy of phonological content processing; the aspects of joint training of segmentals and suprasegmentals; the typical feedback issues which arise in the CAPT systems; and, finally, some directions for future work based on theories drawn from cognitive linguistics and second language acquisition.

## 2. Literature Review

Pronunciation teaching exploits the potential of speech processing to individualize and adapt the learning of pronunciation. The underlying idea is to build CAPT systems which are able to automatically identify the differences between learner production and a model, and provide an appropriate user feedback. Comparing speech sample to an ideal reference is acknowledged to be a more consistent basis for scoring instead of general statistical values such as mean, standard deviation or probability density distribution of learner's speech records [25,26,30]. Immediate actionable feedback is paramount [7,31,32]. Learner analysis of feedback can significantly improve pronunciation and has been used on various pronunciation features (e.g., segmentals [33,34]; suprasegmentals [26,35]; vowels [36]).

CAPT systems frequently include signal processing algorithms and automated speech recognition (ASR), coupled with specific software for learning management, teacher-student communication, scheduling, grading, etc [37]. Based on the importance of the system-building technology incorporated, the existing CAPT systems can be grouped into four classes namely: visual simulation, game playing, comparative phonetics approach, and artificial neural network/machine learning [28]. ASR incorporating neural networks, probabilistic classifiers and decision making schemes, such as GMM, HMM and SVM, is one of the most effective technologies for CAPT design. Yet, some discrete precautions should be taken to apply ASR in a controlled CAPT environment [38]. A case in point is the

unsuitability of ASR for automatic evaluation of learner input. Instead, ASR algorithms are harnessed in CAPT to obtain the significant acoustic correlates of speech to create a variety of pronunciation learning content, contextualized in a particular speech situation [38]. Pronunciation teaching covers both segmental and suprasegmental aspects of speech. In natural speech, tonal and temporal prosodic properties are co-produced [39]; and, therefore, to characterize and evaluate non-native pronunciation, modern CAPT systems should have the means to collectively represent both segmentals and suprasegmentals [27]. Segmental (phonemic and syllabic) activities work out the correct pronunciation of single phonemes and co-articulation of phonemes into higher phonological units. Suprasegmental (prosodic) pronunciation exercises embrace word and phrasal levels. Segmental features (phonemes) are represented nominally and temporally, while suprasegmental features (intonation, stress, accent, rhythm, etc.) have a large scale of representations including pitch curves, spectrograms and labelling based on a specific prosodic labelling notation (e.g., ToBI [40], IViE [41]). The usability of CAPT tools increases if they are able to display the features of natural connected speech such as elision, assimilation, deletion, juncture, etc. [42].

- At word level the following pronunciation aspects can be trained:
    - stress positioning;
    - stressed/unstressed syllables effects, e.g., vowel reduction; and
    - tone movement.
- Respectively, at phrasal level the learners might observe:
    - sentence accent placement;
    - rhythmic pattern production; and
    - phrasal intonation movements related to communicative functions.

Contrasting the exercises as phonemic/prosodic as well as the definition of prosody purely in terms of suprasegmentals is a question under discussion so far, because prosodic and phonemic effects can naturally co-occur. When prosody is expressed through suprasegmental features one can also observe the specific segmental effects. For example, tone movement within a word shapes the acoustic parameters that influence voicing or articulation [43]. Helping learners understand how segmentals and suprasegmentals collectively work to convey the non-verbal speech payload is one of the most valuable pedagogical contributions of CAPT.

A visual display of the fundamental frequency $f_0$ (which is the main acoustical correlate of stress and intonation) combined with audio feedback (such as it is demonstrated in Figure 1b) is generally helpful. However, the feasibility of visual feedback increases if the learner's $f_0$ contour is displayed not only along with a native model, but with possible formalized interpretation of the difference between the model and the learner. Searching for objective and reliable ways to tailor instantaneous corrective feedback about the prosodic similarity between the model and the learner's pitch input remains one of the most problematic issues of CAPT development and operation [29].

The problem of prosodic similarity evaluation arises within multiple research areas including automatic tone classification [44] and proficiency assessment [45]. Although in [46] it was shown that the difference between two given pitch contours may be perfectly grasped by Pearson correlation coefficient (PCC) and Root Mean Square (RMS), in the last two decades, prosodic similarities were successfully evaluated using dynamic time warping (DTW). In [47], DTW was shown to be effective at capturing the similar intonation patterns being tempo invariant and, therefore, more robust than Euclidean distance. DTW combines the prosodic contour alignment with the ability to operate with prosodic feature vectors containing not only $f_0$, but the whole feature sets, which can be extracted within a specific system. There is also evidence to that learner performance might be quantified by synchronization dynamics between model and learner achieved gradually in the course of learning. Accordingly, the applicable metrics may be constructed using cross recurrence quantification metrics (CRQA metrics) [48,49]. The effects of prosodic

synchronization have been observed and described by CRQA metrics in several cases of emotion recognition [50] and the analysis of informal and business conversations [51].

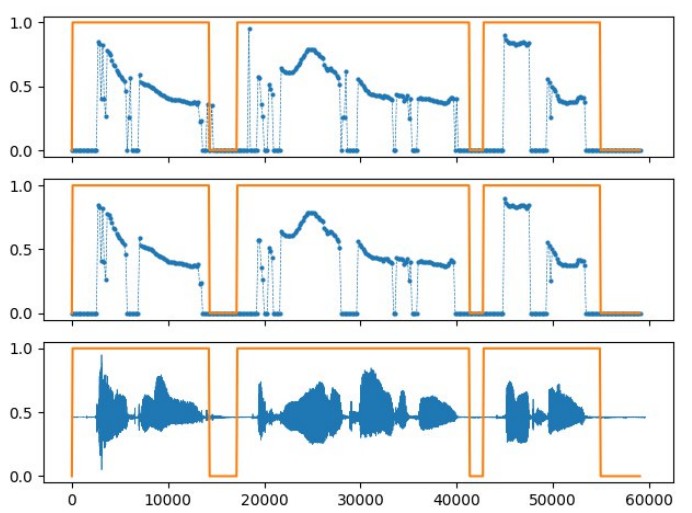 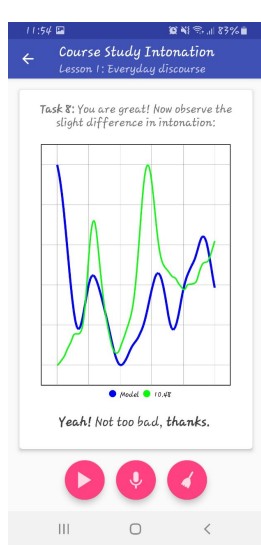

(**a**) Voice activity for task 8 in lesson 1 "Yeah, not too bad, thanks"! (**b**) User screen

**Figure 1.** Voice activity detection process and pitch curves displayed by the mobile application.

Though speech recognition techniques (which are regularly discussed in the literature, e.g., in [52,53]) are beyond the scope of this paper, some state-of-the-art approaches of speech recognition may enhance the accuracy of user speech processing in real time. As such, using deep neural networks for recognition of learner speech might provide a sufficiently accurate result to be used to echo learner's speech input and can be displayed along with the other visuals. The problem of background noise during live sound recording, which is partially solved in the present work by voice activity detection, can be tackled as per the separation of interfering background based on neural algorithms [54].

## 3. Methods

### 3.1. Voice Activity Detection (VAD)

Visual feedback for intonation has become possible due to fundamental frequency (pitch) extraction which is a conventional operation in acoustic signal processing. Pitch detection algorithms output the pitches as three vectors of *time* (timestamps), *pitch* (pitch values) and *conf* (confidence). However, the detection noise, clouds and discontinuities of pitch points, and sporadic prominences make the raw pitch series almost indiscernible by the human eye and unsuitable to be visualized "as is". Pitch detection, filtering, approximation and smoothing are standard processing stages to obtain a pitch curve adapted for teaching purposes. When live speech recording for pitch extraction is assumed, a conventional practice is to apply some VAD to raw pitch readings before the other signal conditioning stages. To remove possible recording imperfections, VAD was incorporated into StudyIntonation DSPCore. VAD was implemented via a three-step algorithm as per [55] with *logmmse* [56] applied at the speech enhancement stage. In Figure 1a, the upper plot shows the raw pitch after preliminary thresholding in a range between 75 Hz and 500 Hz and at a confidence of 0.5. The plot in the middle shows the pitch curve after VAD. The last plot is speech signal in time. The rectangle areas on all plots mark the signal intervals where voice has been detected. Pitch samples outside voiced segments are suppressed, the background noise before and after the utterance as well as hesitation pauses are excluded. The mobile app GUI is shown in Figure 1b.

### 3.2. Prosodic Similarity Evaluation

In its application to the domain of learner's speech production evaluation, DTW accounts for prosodic similarity tolerably well. There are, however, two inherent limitations: firstly, DTW score is static and produces only an instant snap-shot of learner's performance; and secondly, it is too holistic and general: the DTW score gives only one numerical value to a learner, and this value is hardly interpretable in terms of what to undertake to succeed or improve. On the other hand, the effects of synchronization and coupling that may be provided with the help of CRQA are more promising for constructing and representing more comprehensible CAPT feedback.

As soon as the learner is expected to synchronize the prosodic characteristics to a certain extent with the model, CRQA metrics might provide an insight into learning dynamics or, at least, have stronger correlation with "good" or "bad" attempts of a learner. Consequently, it might be reasonable to examine whether synchronization phenomenon occurs, register CRQA metrics and apply them collectively with DTW.

Natural processes can reveal recurrent behavior, e.g., periodicities and quasi-regular cycles [48]. Recurrence of states, meaning the states converging to become arbitrary close after sometime, is a fundamental property of deterministic dynamical systems and is typical for nonlinear or chaotic systems [57,58]. Recurrences in the dynamics of a dynamical system can be visualised by the recurrence plot (RP), which represents the times at which the states $x_i$ recur.

Recurrence of a state at time $i$ at another time $j$ is plotted on a two-dimensional squared matrix $R$ with dots, where both axes represent time:

$$R_{i,j}^{m,\epsilon} = \Theta(\epsilon_i - ||x_i - x_j||), x_i \in R_m, i, j = 1..N,\tag{1}$$

where $N$ is the number of considered states of a model $x_i$ and a learner $x_j$; $\epsilon$ is a threshold distance; $\Theta(x)$ is the Heaviside function.

To quantify the recurrence inside a system a set of recurrence variables may be defined [48]. In the present study two are used: recurrence rate ($RR$) and percent determinism ($DET$).

$RR$ is a measure of the relative density of recurrence points; it is related to the correlation sum:

$$RR(\epsilon, N) = \frac{1}{N^2 - N} \sum_{i \neq j=1}^{N} R_{i,j}^{m,\epsilon}\tag{2}$$

$DD$ describes the density of RP line structures. $DD$ is calculated via $H_D(l)$—the histogram of the lengths of RP diagonal structures:

$$H_D(l) = \sum_{i \neq j=1}^{N} (1 - R_{i-1,j-1})(1 - R_{i+1,j+1}) \prod_{k=0}^{l-1} R_{i+k,j+k}\tag{3}$$

Hence, $DD$ is defined as the fraction of recurrence points that form the diagonal lines:

$$DD = \frac{\sum_{l=d_{min}}^{N} l H_D(l)}{\sum_{i \neq j=1}^{N} R_{i,j}^{m,\epsilon}}\tag{4}$$

$RR$ is connected with correlation between model and learner, while $DD$ values show to what extent the learners production of a given task is stable.

### 3.3. Phoneme Processing and Transcription

Phoneme and rhythm processing are performed by the automatic speech recognition component of DSPCore which incorporates the Kaldi ASR tool with the pretrained LibriSpeech ASR model. Kaldi [59] is a full-fledged open source speech recognition tool, where the results might be obtained at any intermediary stage of speech to text process-

ing. At the moment, there is a wide variety of ASR models, suitable for use in the Kaldi library. For English speech recognition, the most popular ASR models are ASpIRE Chain Model and LibriSpeech ASR. These models were compared in terms of vocabulary size and recognition accuracy. The ASpIRE Chain Model Dictionary contains 42,154 words, while the LibriSpeech Dictionary contains only 20,006 words, but has a lower error rate (8.92% against 15.50% for the ASpIRE Chain Model). The LibriSpeech model was chosen for the mobile platform to use with Kaldi ASR due to its compactness and lower error rate.

The incoming audio signal is split into equal frames of 20 ms to 40 ms, since the sound becomes less homogeneous at a longer duration with a respective decrease in accuracy of the allocated characteristics. The frame length is set to a Kaldi optimal value of 25 ms with a frame offset of 10 ms. Acoustic signal features including mel-frequency cepstral coefficients (MFCC), frame energy and pitch readings are extracted independently for each frame (Figure 2). The decoding graph was built using the pretrained LibriSpeech ASR model. The essence of this stage is the construction of the *HCLG* graph, which is a composition of the graphs H, C, L, G:

- Graph *H* contains the definition of HMM. The input symbols are parameters of the transition matrix. Output symbols are context-dependent phonemes, namely a window of phonemes with an indication of the length of this window and the location of the central element.
- Graph *C* illustrates context dependence. The input symbols are context-dependent phonemes, the output symbols of this graph are phonemes.
- Graph *L* is a dictionary whose input symbols are phonemes and output symbols are words.
- Graph *G* is the graph encoding the grammar of the language model.

Having a decoding graph and acoustic model as well as the extracted features of the incoming audio frames, Kaldi performs lattice definitions, which form an array of phoneme sets indicating the probabilities that the selected set of phonemes matches the speech signal. Accordingly, after obtaining the lattice, the most probable phoneme is selected from the sets of phonemes. Finally, the following information about each speech frame is generated and stored in a .ctm file:

- the unique audio recording identifier
- channel number (as all audio recordings are single-channel, the channel number is 1)
- timestamps of the beginning of phonemes in seconds
- phoneme duration in seconds
- unique phoneme identifier

Using the LibriSpeech phoneme list, the strings of the .ctm file are matched with the corresponding phoneme unique identifier.

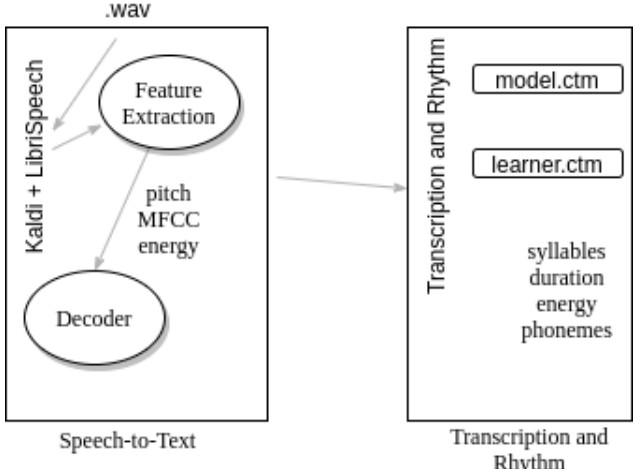

**Figure 2.** Sound processing workflow within DSPCore.

### 3.4. Rhythmic Pattern Retrieval

Kaldi output transcription of text at the phonemic level was further used for splitting the source text into syllables, finding their lengths, highlighting pauses and constructing the rhythm. The set of phonemes in the LibriSpeech model has 4 postfixes: "B" (beginning), "I" (internal), "E"(ending), and "S" (single). This feature of LibriSpeech phonemes allows individual words from a sequence of phonemes to be selected and split into syllables.

For most English words, the number of syllables is equivalent to the number of vowel phonemes. This rule will be used further since due to phonological peculiarities of the English language there is no single algorithm for syllable splitting. Two possible ways could be using a dictionary of English words with manual syllabic segmentation or exploiting language properties that allow automatic segmentation with a certain error percentage. Syllabic segmentation is based on the information about word boundaries and vowels in a word, and also adhere to the rule of maximum number of consonants at the beginning of a syllable [38]. This method achieved segmentation accuracy of 93%. It is possible to increase the accuracy by adding manually selected words that do not lend themselves to the common rule. At this stage, it is also necessary to take into account the pauses between syllables and syllable duration for the subsequent rhythmic pattern retrieval. We store the data of syllables and pauses in the following data structure: data type; syllable or pause; duration; start time; maximum energy on a data interval; and an array of phonemes included. In the case of a pause the maximum energy is taken to be zero, and the array of phonemes contains the phoneme, defining silence, labeled SIL in LibriSpeech. The duration of a syllable or pause is calculated as the sum of all phoneme durations of a syllable. Relative start time is assumed as the start time of the first phoneme in the syllable.

The maximum energy is calculated using MFCC obtained for all frames of the audio signal (Section 3.3). Accordingly, to find the maximum energy, it is necessary to conjoin the frames related to a given syllable and take the maximum entry of the frame energy in this area. Knowing the start and end timestamps of a syllable, one can get the first frame containing the beginning of the syllable and the last frame containing the last phoneme. This operation is linear to the number of syllable frames. The obtained syllable characteristics are used to mark the stressed and unstressed syllables. It is also worth noting that LibriSpeech uses AB notation to represent phonemes, for this reason, before writing the results in json format it is necessary to carry out the conversion of phonemes to IPA1 notation. All this information (Table 1, Figure 3a,b) is used to construct visualization and feedback on rhythm, which can be displayed either jointly with phonetic transcription or separately in the form of energy/duration rectangular patterns (Figure 4a,b). In case, when transcription and rhythm are addressed separately, the former might be visualized as aligned orthographic and phonetic phrases (Figure 5). Technically, all ASR-based features are implemented as the following C++ software components:

- Transcription API
- Transcription Recognizer
- Transcription Analyzer
- Syllable Builder

**Table 1.** Transcription and energy for task 8 lesson 1.

| "Yeah, Not too Bad, Thanks"! | | | | | |
| --- | --- | --- | --- | --- | --- |
| Energy | 27.3 | 24.2 | 18.1 | 30.8 | 24.6 |
| Model | ɛ | nɑːt | tiː | bæd | θæŋks |
| Learner | oʊ | nɑːt | tuː | bɛd | θæŋks |

```
{
    "uttID": "utt3",
    "phonologicalObjects": [{
        "type": "pause",
        "duration": 0.49000000953674319,
        "startTime": 0.0,
        "energy": 0,
        "phones": []
    }, {
        "type": "syllable",
        "duration": 0.25999999046325686,
        "startTime": 0.49000000953674319,
        "energy": 27.337,
        "phones": "ɛ"
    }, {
        "type": "pause",
        "duration": 0.05000000074505806,
        "startTime": 0.75,
        "energy": 0,
        "phones": []
    }, {
        "type": "syllable",
        "duration": 0.2199999988079071,
        "startTime": 0.800000011920929,
        "energy": 24.2176,
        "phones": "nɑːt"
    }, {
        "type": "syllable",
        "duration": 0.12000000476837158,
        "startTime": 1.0199999809265137,
        "energy": 18.0762,
        "phones": "tiː"
    }, {
```
```
    }, {
        "type": "syllable",
        "duration": 0.12000000476837158,
        "startTime": 1.0199999809265137,
        "energy": 18.0762,
        "phones": "tiː"
    }, {
        "type": "syllable",
        "duration": 0.30000001192092898,
        "startTime": 1.1399999856948853,
        "energy": 30.8128,
        "phones": "bæd"
    }, {
        "type": "pause",
        "duration": 0.009999999776482582,
        "startTime": 1.440000057220459,
        "energy": 0,
        "phones": []
    }, {
        "type": "syllable",
        "duration": 0.5399999618530273,
        "startTime": 1.4500000476837159,
        "energy": 24.5894,
        "phones": "θænks"
    }, {
        "type": "pause",
        "duration": 0.07999999821186066,
        "startTime": 1.9900000095367432,
        "energy": 0,
        "phones": []
    }]
}
```

(**a**)                                                            (**b**)

**Figure 3.** .ctm file for task1 in lesson 1 "Yeah, not too bad, thanks!". The syllabic partition for "Yeah, not too..." is shown in (**a**) and the rest of the phrase, "... bad, thanks", is presented in (**b**).

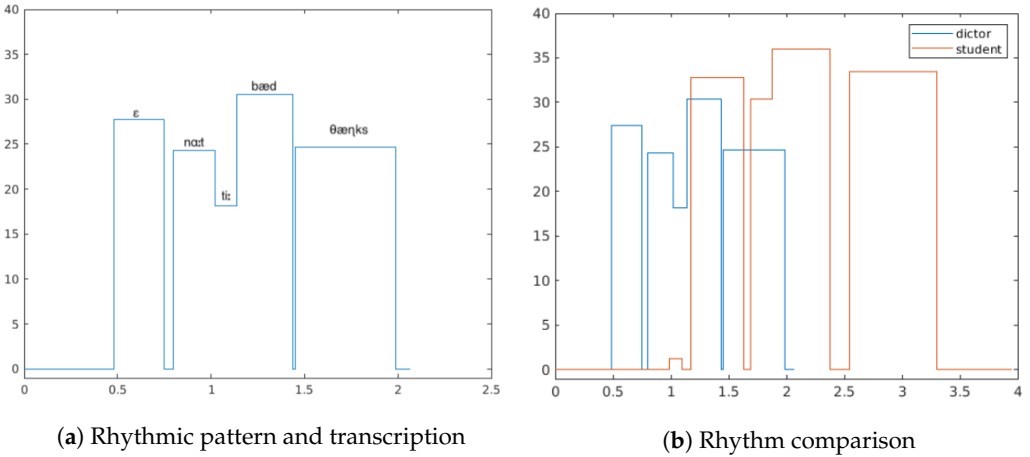

(**a**) Rhythmic pattern and transcription                (**b**) Rhythm comparison

**Figure 4.** Visual feedback on rhythm is produced with joint representation of phonetic transcription and energy/duration rectangular pattern (**a**). Model and learner patterns can be obtained and shown together (**b**). In this case, both transcriptions are not shown to eliminate text overlapping.

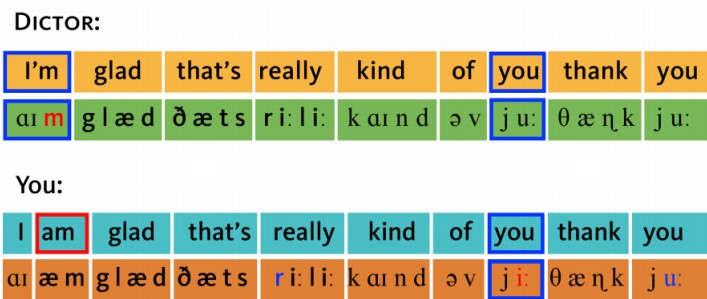

**Figure 5.** A prototype of user interface to show phonemic distortions in task 25 lesson 1.

### 3.5. Content Structure

StudyIntonation courseware is organized using a hierarchical phonological structure to display the distribution of suprasegmental features and how they consequently influence the timing of phones and syllables; thus, defining variation in their phonetic implementation. For example, the fundamental frequency F0 in English is a correlate of phrasal prominence and associates with pitch accent increased duration which is a consistent correlate of stress.

According to [43], the phonological representation is layered (Figure 6), and the elements at a lower level combine to form higher level prosodic constituents such as words and phrases (Figure 7). Using this hierarchical approach, prosody is represented in terms of boundaries that mark the edges and prominences such as stress and accent which are assigned to an element within the prosodic constituent at a given level [43].

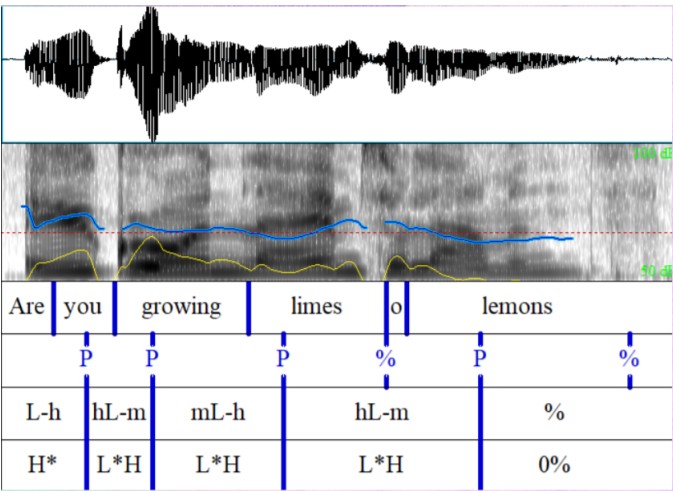

**Figure 6.** Praat window with multi-layer IViE labelling of the utterance [Are you growing limes][or lemons?].

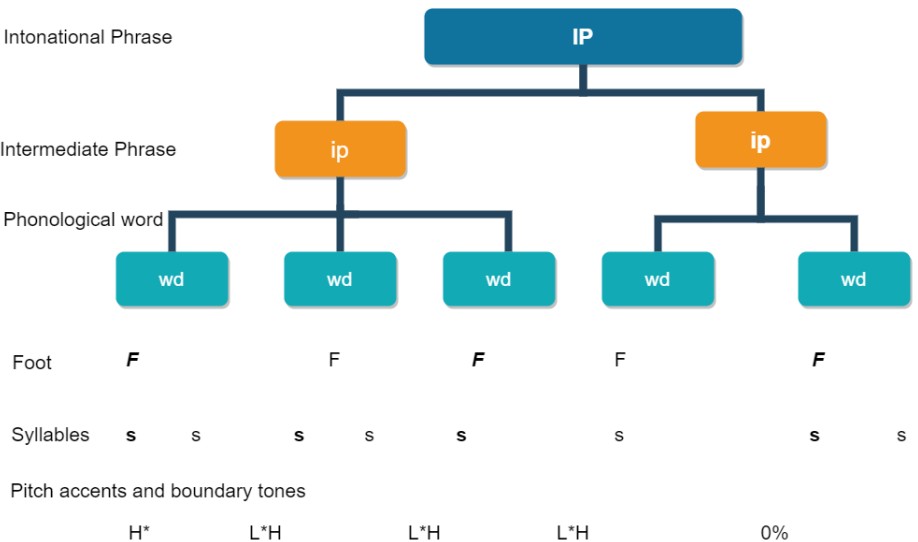

**Figure 7.** Diagram of an utterance [Are you growing limes][or lemons?] showing hierarchically layered prosodic structure at the syllable, foot, word and phrase levels. Prominent positions are given in boldface.

## 4. Results

### 4.1. DSPCore Architecture

Digital signal processing software core (DSPCore) of StudyIntonation has been designed to process and display model and learner pitch curves and calculate some proximity metric-like correlation, mean square error and dynamic time warping distance. However, this functionality covers only the highest level of pronunciation teaching: the phrasal intonation. With regard to providing a display of more phonological aspects of pronunciation such as phonemes, stress and rhythm and in favour to a holistic approach to CAPT, we integrated an ASR system Kaldi with the pre-trained ASR-model Librispeech into the DSPCore (Figure 2). Thus, in addition to the pitch processing thread, which was instrumented with VAD block, three other threads appeared as the outputs of ASR block: (1) for phonemes; (2) for syllables, stress and rhythm; and (3) for recognised words (Figure 8). All these features are currently produced offline for model speech and will be incorporated into mobile CAPT to process learners speech online.

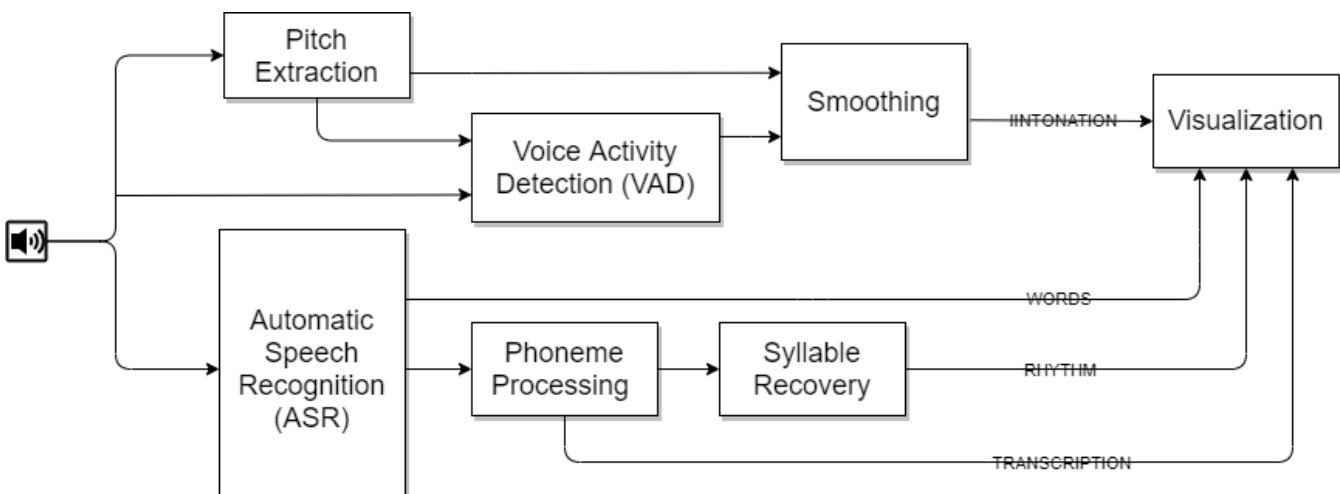

**Figure 8.** StudyIntonation Signal Processing Core (DSPCore).

### 4.2. DSPCore Assessment with Reference Corpora

Transcription module code has been validated using phonemic labelling of TIMIT and tonal and prominence labelling of IViE corpora. Labelling for both corpora was performed by human experts to provide a representation of the ground truth for phonemic, tonal and prominence content of the data.

To assess the recognition quality, the distance metric was calculated as Levenstein distance $L$, which shows how much two phoneme sequences differ [60]. Metric for sequences of equal length (e.g., Hamming distance) cannot be used in this task, because two sequences can differ in their length due to possible substitutions, deletions, or insertions of phonemes.

After receiving a phonemic dictionary for translation into IPA1 notation, the module for recognizing and constructing phonetic transcription was applied to a set of phrases from the TIMIT corpus, which bears a reference phonemic labelling. For translation into IPA1 notation we used Praat to overlay the corpus record orthography with its phonetic transcription and translate it into IPA1 notation (Figure 9).

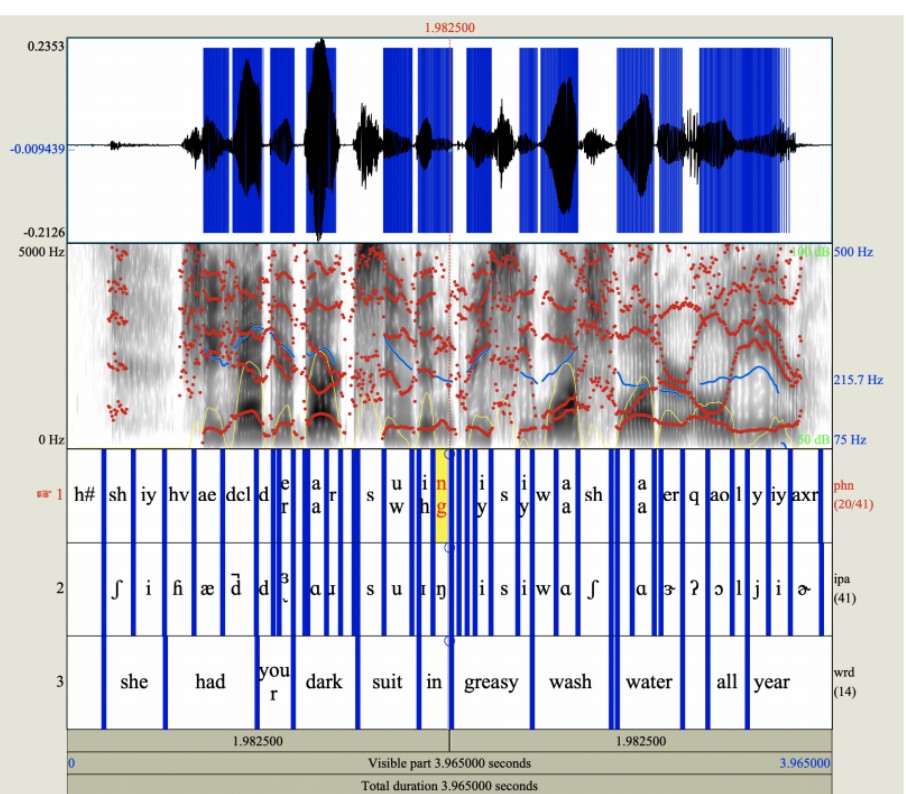

**Figure 9.** AB to IPA1 translation.

The algorithm for calculating the Levenstein distance consists of constructing a matrix $D$ of dimension $m \times n$, where $m$ and $n$ are lengths of sequences, and calculating the value located in the lower right corner of $D$:

$$D(i,j) = \begin{cases} 0 & i = 0, j = 0 \\ i & i > 0, j = 0 \\ j & i = 0, j > 0 \\ min\{D(i, j-1) + 1, D(i-1, j) + 1, D(i-1, j-1) + m(S_1[i], S_2[j])\} & i > 0, j > 0 \end{cases}$$

where

$$m(a,b) = \begin{cases} 0 & a = b \\ 1 & a \neq b \end{cases}$$

and, finally,

$$L(S_1, S_2) = \frac{D(S_1, S_2)}{max(|S_1|, |S_2|)}.$$

Table 2 shows transcriptions for a stimulus phrase from TIMIT corpus, one made by human experts (TIMIT row) and one obtained from DSPCore (DSPCore row). It can be observed that the vowels and consonants are recognised correctly, yet, in some cases, DSPCore produces a transcription which is more "correct" from the point of view of Standard English pronunciation, because automatic transcription involves not only acoustic signal, but the language model of ASR system as well.

The normalised value of *L* oscillates in the interval between 0 and 1. Table 3 shows that the typical values for TIMIT are in a range of 0.06–0.28 (up to 30% of phonemes are recognized differently by DSPCore). Among the differences in phrases, one can note the absence of the phoneme "t" of the word "don't". This difference stems from the peculiarities of elision, that is, "swallowing" the phoneme "t" in colloquial speech. At the same time, the recognition module recorded this phoneme, which, of course, is true from the point of view of constructing transcription from the text; however, it is incorrect for colloquial speech. A similar conclusion can be reached regarding the phoneme "t" in the word "that". Another case is the mismatched vowel phoneme ə with the vowel phoneme of the ɪ of TIMIT. Phonemes ə and ɪ have similar articulation, so they might have very close decoding weights. Among the notable differences in this comparison, one can single out the absence of elision of vowel and consonant phonemes in the transcription resulting from the recognition. Overall, the resulting transcription matches the original text more closely than the transcription given in the corpus. However, at the same time, the transcription from the TIMIT corpus is a transcription of colloquial speech, which implies various phonetic distortions. Many assumptions in the pronunciation of colloquial words impinged on recognition accuracy.

**Table 2.** Transcription details.

| Phrase 2 | |
|---|---|
| Text | **don't ask me to carry an oily rag like that** |
| TIMIT | do ʊn æsk miː tɪ kɛriː m ɔɪliː ræg lɑɪk ðæ |
| DSPCore | do ʊnt æsk miː tɪ kɛriː ən ɔɪliː ræg lɑɪk ðæt |
| **Phrase 4** | |
| Text | **his captain was thin and haggard and his beautiful boots were worn and shabby** |
| TIMIT | hɪz kætɪn wəs θɪn æn hægɛːrd ɪn ɪz bjʊtʊfl buːts wr wɔːrn ɪn ʃæbiː |
| DSPCore | hɪz kæptən wəz θɪn ənd hægɜːrd ænd hɪz bjuːtəfəl buːts wɜːr wɔːrn ænd ʃæbiː |
| **Phrase 10** | |
| Text | **drop five forms in the box before you go out** |
| TIMIT | drɑː fɑɪv fɔːrmz ən ðə bɑːks bəfːr jɪ gʊ aʊ |
| DSPCore | drɑːp fɑɪv fɔːrmz ɪn ðə bɑːks bɪfɔːr juː gʊ aʊt |

**Table 3.** Examples of Levenstein distance between the reference transcription of TIMIT and the one obtained by DSPCore transcription module.

| Number | TIMIT Phrase | *L* |
|---|---|---|
| 1 | She had your dark suit in greasy wash water all year. | 0.16 |
| 2 | Don't ask me to carry an oily rag like that. | 0.06 |
| 3 | Production may fall far below expectations. | 0.20 |
| 4 | His captain was thin and haggard and his beautiful boots were worn and shabby. | 0.28 |
| 5 | Her wardrobe consists of only skirts and blouses. | 0.15 |
| 6 | The reasons for this dive seemed foolish now. | 0.16 |
| 7 | Elderly people are often excluded. | 0.15 |
| 8 | Pizzerias are convenient for a quick lunch. | 0.23 |
| 9 | Put the butcher block table in the garage. | 0.20 |
| 10 | Drop five forms in the box before you go out. | 0.13 |

### 4.3. Prosodic Similarity Evaluation Assessment

As it is mentioned earlier in this article, currently, despite the drawbacks related to tailored CAPT feedback production, DTW may be used as a conventional algorithm to calculate the distance between two time series. This algorithm is particularly useful if the signals have a similar shape but different argument scale. While evaluating prosodic similarity, this issue may occur in the case of different speech rates of the model and the learner.

To verify our DTW scoring algorithm in StudyIntonation, we performed a try-out with Praat and NumPY. A group of teachers and students from our universities (including 1 British, 1 Canadian, 2 Japanese, 1 Chinese and 1 Russian), were asked to record multiple attempts of twenty-two IViE stimuli sentences. The records were processed by StudyIntonation so that we can get $f_0$ readings and calculate DTW; at the same time, these records were processed with Praat and NumPY. We calculated not only DTW "model-speaker", but also "speaker-speaker" for each sentence. The results for "speaker-speaker" DTW score for one sentence are given in Table 4. DTW scores for one participant are given in boldface to show the possible maximum values of DTW score.

Table 5 contains DTW readings, averaged over attempts, for 12 arbitrary sentences from IViE for a native speaker (referred as B–B) and for participants with L1 Russian (R–R), Japanese (J–J), Chinese (C–C), respectively. Averaged pairwise DTW scores between native and non-native speakers are referred as R–B, J–B, C–B. The last column $< DTW >$ is average DTW score over all sentences and all attempts for a speaker.

**Table 4.** Pairwise DTW "speaker-speaker" for one attempt.

|  | B-1 | J-f2 | J-1 | C-2 | C-1 | J-2 | R-2 | J-f1 | C-1 | R-1 |
|---|---|---|---|---|---|---|---|---|---|---|
| **B-1** | 0 | 7.79315 | 9.82649 | 11.3357 | 10.6621 | 8.77679 | 5.08777 | **29.6911** | 3.33005 | 6.45209 |
| **J-f2** | 7.79315 | 0 | 5.41653 | 11.4249 | 10.5547 | 7.91541 | 5.80402 | **29.2233** | 7.58529 | 6.7341 |
| **J-1** | 9.82649 | 5.41653 | 0 | 12.3131 | 12.2374 | 8.64963 | 7.48466 | **19.1596** | 10.4463 | 9.83542 |
| **C-2** | 11.3357 | 11.4249 | 12.3131 | 0 | 2.72298 | 6.24624 | 16.1561 | **42.3357** | 10.1632 | 14.6927 |
| **C-1** | 10.6621 | 10.5547 | 12.2374 | 2.72298 | 0 | 6.8801 | 16.3171 | **44.3193** | 9.53339 | 13.0754 |
| **J-2** | 8.77679 | 7.91541 | 8.64963 | 6.24624 | 6.8801 | 0 | 11.0279 | **35.6397** | 6.62749 | 7.60256 |
| **R-2** | 5.08777 | 5.80402 | 7.48466 | 16.1561 | 16.3171 | 11.0279 | 0 | **18.9842** | 4.71101 | 3.76105 |
| **J-f1** | **29.6911** | **29.2233** | **19.1596** | **42.3357** | **44.3193** | **35.6397** | **18.9842** | 0 | **27.52** | **16.5104** |
| **C-1** | 3.33005 | 7.58529 | 10.4463 | 10.1632 | 9.53339 | 6.62749 | 4.71101 | **27.52** | 0 | 3.84188 |
| **R-1** | 6.45209 | 6.7341 | 9.83542 | 14.6927 | 13.0754 | 7.60256 | 3.76105 | **16.5104** | 3.84188 | 0 |

**Table 5.** Average individual and pairwise DTW scores.

| Sentence | 1 | 3 | 6 | 7 | 9 | 12 | 15 | 17 | 18 | 19 | 20 | 21 | $< DTW >$ |
|----------|------|------|------|------|------|------|------|------|------|------|------|------|------|
| B–B | 3.33 | 1.7 | 0.55 | 1.43 | 0.98 | 0.88 | 4.83 | 2.06 | 1.82 | 2.89 | 2.57 | 0.49 | **2.32** |
| R–R | 3.76 | 0.73 | 3.16 | 1.38 | 3.58 | 2.33 | 2.06 | 3.62 | 2.01 | 2.06 | 5.04 | 4.5 | **3.02** |
| J–J | 29.22 | 8.51 | 1.4 | 5.29 | 3.58 | 44.3 | 2.96 | 1.5 | 1.47 | 8.02 | 4,99 | 0.54 | **9.43** |
| C–C | 2.72 | 1.95 | 2.63 | 1.35 | 2.59 | 1.41 | 1.22 | 1.13 | 0.46 | 1.31 | 1.87 | 1.79 | **1.12** |
| R–B | 5.08 | 2.38 | 5.69 | 8.5 | 1.63 | 5.31 | 5.99 | 2.78 | 2.16 | 4.51 | 3.0 | 1.7 | **7.1** |
| J–B | 7.79 | 5.02 | 1.55 | 2.13 | 7.58 | 2.01 | 1.31 | 4.5 | 1.76 | 1.81 | 2.12 | 2.20 | **6.7** |
| C–B | 10.66 | 10.2 | 1.68 | 0.38 | 6.64 | 5.17 | 3.73 | 5.33 | 3.13 | 10.7 | 6.28 | 2.68 | **6.2** |

Though DTW metrics are helpful to estimate how close the user's pitch is to the model, these metrics are not free both from false positives and negatives. Table 5 last column shows the average DTW scores between several attempts of one speaker who pronounced the same phrase. For native speakers these scores are relatively small: 2.32 for British speaker, 1.12 for Canadian speaker, Russian speaker with a good knowledge of English had a score of 3.02. However, in general, there is much oscillation in *DTW* scores.

We searched for more stable scoring method which could possibly give an insight into the learners' dynamics by the analysis of model and learner co-trajectory recurrence, so the CRQA metrics *RR* and *DD* were calculated. The main advantage of CRQA is that it makes possible to analyse the dynamics of a complex multi-dimensional system using only one system parameter, namely $f_0$ readings. The present research release of StudyIntonation outputs model and learner $f_0$ series to an external NumPY module, which can obtain *RR* and *DD* while the learner is training.

We trained two binary classifiers, namely logistic regression and decision tree of depth 2 and minimal split limit equal to 5 samples, to find out whether one can get a cue regarding learner performance using *DTW RR* and *DD* and what combination returns a better accuracy of classification. The dataset was formed by multiple attempts of 74 StudyIntonation tasks recorded from 7 different speakers (1 native speaker, 6 non-native speakers whose first language is Russian). Each attempt was labelled by a human expert as "good" and "poor", *DTW RR* and *DD* metrics were calculated between a model from the corresponding task and the speaker. To estimate the discriminative ability of *RR* and *DD* we performed 5-fold cross-validation, which results are shown in Table 6.

**Table 6.** Discriminating ability of DTW and CRQA feature combinations.

| Accuracy on a Test Set | DTW | RR | DD | DTW + RR | DTW + DD | RR + DD | DTW + RR + DD |
|---|---|---|---|---|---|---|---|
| Logistic Regression | **0.68** | 0.60 | 0.59 | **0.75** | 0.71 | 0.64 | 0.76 |
| Decision Tree | 0.61 | **0.88** | 0.73 | **0.87** | 0.75 | 0.81 | **0.87** |
| **Feature Importance** | | | | | | | |
| DTW | 1.00 | - | - | 0.12 | 0.10 | - | 0.09 |
| RR | - | 1.00 | - | **0.88** | - | **0.94** | **0.88** |
| DD | - | - | 1.00 | - | 0.06 | 0.06 | 0.03 |

The feature space in Figure 10 shows a stronger discriminating ability of *RR* metric in comparison with *DTW*. In the current experiment the application of both *RR* and *DD* together with *DTW* does not add much to the accuracy because of a strong correlation between these CRQA metrics. Therefore, any of the CRQA metrics (either *RR* or *DD*) can be used for better attempt classification.

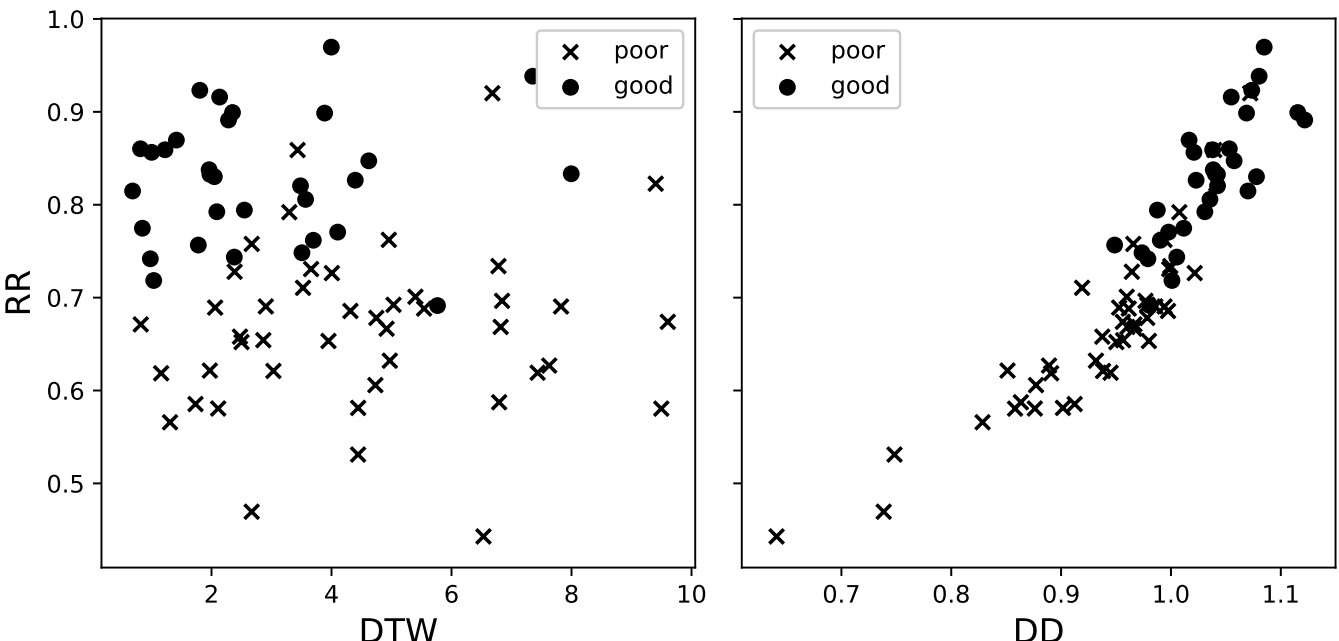

**Figure 10.** Single task attempts in feature space: DTW + RR (**left**), DD + RR (**right**).

### 4.4. User Try-Outs with StudyIntonation Courseware

In addition to professional assessment, it is important to analyze the end users' experience. That is why, to evaluate how the learning tasks are processed by DSPCore, we involved several language learners (5 Russian learners with upper-intermediate level of English, in our case). They performed 74 StudyIntonation tasks grouped in 4 lessons. Table 7 illustrates a sample output of these experiments.

**Table 7.** Tasks processed by DSPCore for model and learner.

| | **I've had to cancel the meeting** |
|---|---|
| Model | ɑɪv hæd tə kænsəl ðə mitɪŋ |
| Learner | ɑɪv hɛd tə kænsəl ðə miːtɪŋ |
| | **Good morning. Did you sleep well?** |
| Model | gʊd mɔːrnɪŋ dɪd júː sliːp wɛl |
| Learner | gʊd mɔːrnɪŋ dɪd júː sliːp wɛl |
| | **I know we haven't met before but could I have a word with you please?** |
| Model | ɑɪ noʊ wiː hævənt mɛt bɪfɔːr bət kʊd ɑɪ hæv ə wɜːrd wɪθ júː pliːz |
| Learner | ɑɪ noʊ wiː hævənt mɛt bɪfɔːr bət kʊd ə hæv ə wɜːrd wɪθ júː pleɪs |

Learner and model utterance comparison for feedback is formed using the following ASR internal parameters:

- uttName—unique identifier of the phrase
- utteranc_error_rate—the number of words of the phrase in which mistakes were made
- start_delay_dict—silence before the first word of the speaker
- start_delay_user—silence before the first word of the user
- words—json element of word analysis results
- extra_words—extra words in the user's phrase
- missing_words—missing words in the speaker's phrase

- similar_words—json element of the results of analysis of similar words and their phonemes
- word_content_dictor—phonetic transcription of the speaker's word
- word_content_user—phonetic transcription of the user's word
- start_time_dictor, end_time_dictor, start_time_user, end_time_user—timestamps for the words of the speaker and the user
- extra_phonemes—extra phonemes in the user's word
- missing_phonemes—missing phonemes in the speaker's word
- equal_phonemes—json-element of identical phonemes, created according to logic similar to the logic of creation similar_words
- abnormal_duration—flag of the presence of an abnormal duration of the user's phoneme compared to the duration of the speaker's phoneme
- delta—the absolute difference between the duration of the user's phoneme and output when anomalous duration flag is present

## 5. Discussion

The field of second language (L2) acquisition has seen an increasing interest in pronunciation research and its application to language teaching with the help of present day technology based on signal processing algorithms. Particularly, an idea to make use of visual input to extend the audio perception process has become easier to implement using mobile personalized solutions harnessing the power of modern portable devices which have become much more than simply communication tools. It has led to language learning environments which widely incorporate speech technologies [5], visualization and personalizing based on mobile software and hardware designs [19,61], which include our own efforts [21,22,62].

Many language teachers use transcription and various methods of notation to show how a sentence should be pronounced. The complexity of notation ranges from simple directional arrows or pitch waves to marking multiple suprasegmental features. Transcription often takes the form of English phonemics popularized by [63]. This involves students learning IPA symbols and leads them to acquire a skill of phonetic reading, which consequently corrects some habitual mistakes, e.g., voiced ð and voiceless θ mispronunciation or weak form elimination. Using transcription, connected speech events can be explained, the connected speech phenomena of a model speech may be displayed and the difference between phonemes of a model and a learner may garner hints on how to speak in a foreign language more naturally.

During assessment, DSPCore allowed inaccuracies in the construction of phonetic transcription of colloquial speech. To the best of our knowledge, the cause of these inaccuracies stems from the ASR model used (e.g., Librispeech), which is trained on audio-books performed by professional actors. These texts are assumed to be recorded in Standard British English. That explains why their phonetic transcription is more consistent with the transcription of the text itself rather than with colloquial speech (Table 2). This is also shown by a relatively good accuracy of phrase No. 2 ($L = 0.06$), in which there was a minimum amount of elision of sounds from all the phrases given. For the case when connected speech effects are the CAPT goal, another ASR model instead of Librispeech for ASR decoder training should be chosen or designed deliberately to recognize and transcribe connected speech.

For teaching purposes, when the content is formed with model utterances, carefully pronounced in Standard British English [26] and under the assumption that the learner does not produce connected speech phenomena such as elision, etc., we can conclude that DSPCore works sufficiently well for recognition and, hopefully, is able to provide actionable feedback. CAPT systems such as StudyIntonation supporting individual work of students with technology-enriched learning environments which extensively use the multimedia capabilities of portable devices are often considered to be a natural components of the distance learning process. However, distance learning has its own quirks. One problem

commonly faced while implementing a CAPT system is how to establish a relevant and adequate feedback mechanism [29]. This "feedback issue" is manifold. First and most important, the feedback is required so that both the teacher and the learner are able to identify and evaluate the segmental and suprasegmental errors. Second, the feedback is required to evaluate the current progress and to suggest steps for improvement in the system. Third, the teachers are often interested in getting a kind of behavioral feedback from their students including their interests, involvement or engagement (as distance learning tools may lack good ways to deliver such kind of feedback). Finally, there are also usability aspects.

Although StudyIntonation enables provisioning the feedback in the form of visuals and some numeric scores, there are still open issues in our design such as (1) metric adequacy and sensitivity to phonemic, rhythmic and intonational distortions; (2) feedback limitations when learners are not verbally instructed what to do to improve; (3) rigid interface when the graphs are not interactive; and (4) the effect of context which produces multiple prosodic portraits of the same phrase which are difficult to be displayed simultaneously [62]. Mobile CAPT tools are supposed to be used in an unsupervised environment, when the interpretation of pronunciation errors cannot be performed by a human teacher, thus an adequate, unbiased and helpful automatic feedback is desirable. Therefore, we need some sensible metrics to be extracted from the speech, which could be effectively classified. Many studies have demonstrated that communicative interactions share features with complex dynamic systems. In our case, CRQA scores of recurrence rate and percentage of determinism were shown to increase the binary classification accuracy by nearly 20% in case of joint application with DTW scores. The computational complexity of DTW [47] and CRQA [48] is proportional to several samples in $N$, which in the worst case is $\approx O(N^2)$. The results of StudyIntonation technological and didactic assessment using Henrichsen criteria [64] is provided in [65].

The user tryouts showed that despite the correct match between the spoken phonemes of the learner and those of the model, the rhythm and intonation of a learner did not approach the rhythm and intonation of the model. It proves that suprasegmentals are more difficult to be trained. This can be understood as an argument in favour of joint training of phonetic, rhythmic and intonational features towards accurate suprasegmental usage which is acknowledged as a key linguistic component of multimodal communication [66]. As pronunciation learning is no longer understood as blind copying or shadowing of input stimuli but is acknowledged to be cognitively refracted by innate language faculty; cognitive linguistics and embodied cognitive science might be of help to gain insight into human cognition processes of language and speech; which, in turn, might lead to more effective ways of pronunciation teaching and learning.

**Author Contributions:** S.C., A.L., I.L., A.N. and R.S. designed digital signal processing core algorithmic and software components, adopted them to the purposes of the CAPT system, and wrote the code of signal processing. M.L., D.T. and K.V. integrated speech recognition components into the system and wrote the code to support the phonological features of learning content. E.B. and J.B. worked on CAPT system assessment and evaluation with respect to its links to linguistics and language teaching. N.B. and E.P. designed the system architecture and supervised the project. N.B., E.P. and J.B. wrote the article. All authors have read and agreed to the published version of the manuscript.

**Funding:** This research was funded by Japan Society for the Promotion of Science (JSPS), grant 20K00838 "Cross-disciplinary approach to prosody-based automatic speech processing and its application to the computer-assisted language teaching".

**Conflicts of Interest:** The authors declare no conflict of interest.

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
