# Peer review of "Speech Processing for Language Learning: A Practical Approach to Computer-Assisted Pronunciation Teaching"

_electronics, doi:10.3390/electronics10030235_

Round 1

Reviewer 1 Report

This paper presents a computer-assistant pronunciation assessment method based on speech signal processing and pattern recognition, which is useful in practical teaching. However, the technical contribution of the work and the conclusion of the paper are not clearly described.
The method is based on the modification of the existing system, so the paper should focus on what technical improvements have been implemented and provide experiment data to demonstrate the achieved performance. A suggestion is that, in Section 3.2.1, to compare L values of the TIMIT and the DSPCore so that an improvement rate can be provided. Then extend the comparison from sentences to words to show an improved rate in vowels and consonant phonemes. Additional performance evaluation should be considered. 

Author Response

Response to Reviewer 1

Point 1. [...] the technical contribution of the work and the conclusion of the paper are not clearly described.
The method is based on the modification of the existing system, so the paper should focus on what technical improvements have been implemented and provide experiment data to demonstrate the achieved performance.

Response 1. Thanks to this reviewer's comment. We understood that some things presented in the abstract, motivation and concluding parts of our manuscript may not be obvious for every reader, so we have significantly rewritten and improved the clarity of the abstract. We also added the literature review (Section 2) for better positioning of our work in the scope of existing research. Specifically, let us point out that the paper does not only describe the modification of the existing system. Though the project started in around 2017 and the working system under consideration does exist, we consider this contribution as our first attempt to describe the system in a whole, as well as the models it is based on, the achieved results, and the experiments with the actual version in systematic way, thus, advancing our earlier conference publications which focused on discrete aspects of the system. Regarding to the performance evaluation, please see our responses to Points 2-3.

Point 2. A suggestion is that, in Section 3.2.1, to compare L values of the TIMIT and the DSPCore so that an improvement rate can be provided.

Response 2. Although there exist some corpora with automatic labelling, we have none at our disposal at the moment, so we could perform the comparison only with corpora labelled by human experts. In this sense, TIMIT labelling is a sort of "ground truth", which can be hardly improved, because any automatic transcription may only endeavor to approach the accuracy of human labelling. Therefore, we analysed whether the phonemes are defined correctly and are there any alternations, deletions or insertions in our transcription.
In this case the Levenstein distance is a conventional method which is the most objective for the case of phoneme sequences of possibly different lengths. We spotted several issues connected with colloquial nature of TIMIT stimuli records, e.g. dropped "t"s were inserted by DSPCore. The implications are discussed in the article. Briefly, we can conclude that as the teaching audio content is usually free from strong colloquial peculiarities, we may rely upon the transcription we obtain from DSPCore with having in mind the extension of ASR model to incorporate colloquial and connected speech data. Please see Section 4.2 (pp.12-13) for corresponding explanations in the manuscript.

Point 3. Extend the comparison from sentences to words to show an improved rate in vowels and consonant phonemes. Additional performance evaluation should be considered.

Response 3. Our response to this remark is subsumed within Response 2, because Levenstein distance compares basically the vowel and consonant phonemes of 2 sequences.

Reviewer 2 Report

The work titled "Signal Processing and Speech Recognition for Language Learning: A Practical Approach to Computer-Assisted Pronunciation Teaching" presents an improved system for English language learning in the broader context of optimizing workflow and digitizing study material.

The abstract has the right size and the appropriate abstraction level. However, the three types of output (l.9 p.1) are not clearly distinguished.

Regarding the technical content, the results are interesting and well-explained. Still, a few questions remain. Did the authors consider other alternatives for computing the distance between two phoneme sequences besides the Levenshtein distance? Would a tree-based metric be appropriate for the situation? Can a deep neural network be trained to understand subtle differences in phoneme sequences?

The structure of the text is easy to follow but the lack of a section with a review literature is puzzling. Please create a section after the introduction where recent works about signal processing for speech are briefly presented. Also please add to this section the following works for a better understanding of the field:

+ Purwins, Hendrik, et al. "Deep learning for audio signal processing." IEEE Journal of Selected Topics in Signal Processing 13.2 (2019): 206-219.
+ Xia, Yangyang, et al. "Weighted Speech Distortion Losses for Neural-Network-Based Real-Time Speech Enhancement." ICASSP 2020-2020 IEEE International Conference on Acoustics, Speech and Signal Processing (ICASSP). IEEE, 2020.
+ Yu, Dong, Li Deng, and Frank Seide. "The deep tensor neural network with applications to large vocabulary speech recognition." IEEE Transactions on Audio, Speech, and Language Processing 21.2 (2012): 388-396.
+ Drakopoulos, Georgios, and Phivos Mylonas. "Evaluating graph resilience with tensor stack networks: A keras implementation." Neural Computing and Applications (2020): 1-16.
+ Khamparia, Aditya, et al. "Sound classification using convolutional neural network and tensor deep stacking network." IEEE Access 7 (2019): 7717-7727.
+ Deng, Li, et al. "Recent advances in deep learning for speech research at Microsoft." 2013 IEEE International Conference on Acoustics, Speech and Signal Processing. IEEE, 2013.
+ Deng, Li, Geoffrey Hinton, and Brian Kingsbury. "New types of deep neural network learning for speech recognition and related applications: An overview." 2013 IEEE international conference on acoustics, speech and signal processing. IEEE, 2013.
+ Wang, DeLiang, and Jitong Chen. "Supervised speech separation based on deep learning: An overview." IEEE/ACM Transactions on Audio, Speech, and Language Processing 26.10 (2018): 1702-1726.

The English of this work is more than adequate and clearly the authors have placed emphasis on the language. Still, the authors should consider changing many sentences from active to passive voice (e.g. "We redesigned ..." (l.8 p.1) -> "The working prototype was redesigned...") as the latter is more appropriate for a technical text.

Given the above, I recommend this work be accepted with minor revisions.

Author Response

Response to Reviewer 2

Point 1. [In the abstract], the three types of output (l.9 p.1) are not clearly distinguished.

Response 1. In fact, the system produces 4 types of output, namely, (1) numeric scores, based on dynamic time warping (DTW) and cross-recurrence quantification analysis (CRQA); (2) intonation curve; (3) phonetic transcription; and (4) rhythm in terms of energy and duration of syllables. All these outputs are described in subsections of Section 3. Since, with respect to new contents, we significantly revised the abstract, now there is no explicit reference to the types of output in the abstract, which is now focused on methods and functions instead.

Point 2. Did the authors consider other alternatives for computing the distance between two phoneme sequences besides the Levenshtein distance? Would a tree-based metric be appropriate for the situation? Can a deep neural network be trained to understand subtle differences in phoneme sequences?

Response 2. The Levenstein distance is a conventional method which is the most objective for the case of phoneme sequences of possibly different lengths. We provided at p. 12, line 357, a reference to a work of [Do, 2020], which gives a thorough analysis of various phonological distance measures.

Point 3. Please create a section after the introduction where recent works about signal processing for speech are briefly presented.

Response 3. We created the section "Literature Review" (p.4) providing necessary insight into the state-of-the-art studies in the area. For better positioning of our work, we briefly referred to a number of research projects from the adjacent domains (including those from the list of reviewer's recommendations), which may affect future work and further improvements and enhancements of our system.

Point 4. the authors should consider changing many sentences from active to passive voice [...] as the latter is more appropriate for a technical text.

Response 4. With respect to the reviewer's suggestion, we reconsidered a number of certain cases where the passive voice may improve the style (e.g., line 14-15, 25, 113, 117, etc.). However, let us kindly note that, according to modern language studies, many linguists note that the overuse the passive voice in technical literature may make it difficult to differentiate the actual authors' achievements from other works in the domain. Just to avoid such an overuse, we believe that the active voice should be kept in situations where the contents is about presenting our own views and our own results.

Reviewer 3 Report

In the reviewed paper, the authors proposed signal processing and speech recognition for language learning by the application of intonation speech contour detection, approximation, and visualization. In general

1) The abstract does not say what the novelty is. It must be underlined. Moreover, the obtained results should be written here.
2)The introduction is based on 30 bibliography items and most of them are older than 4 years. I cannot agree that this section shows the current state of knowledge, especially in terms of speech recognition using machine learning approaches.
3) Section 2 shows the proposal, but there is no mathematical model, no novelty of the approaches. The authors focused on the application interface, XML, etc. If it is an application proposal, the full background also should be presented. Especially, the novelty against existing solutions should be underlined here.
4) The experimental section does not examine the proposal. There is no proper analysis of components, no comparison with other solutions from the last 3-4 years, and no deep discussion based on statistical analysis about results.

Based on the above comments, I must reject this paper, because of the initial state of the conducted research.

Author Response

Response to Reviewer 3

Point 1. The abstract does not say what the novelty is. It must be underlined. Moreover, the obtained results should be written here.

Response 1. We have revised the abstract so that it clearly addresses the novelty, presents the achieved practical results, and corresponds to the changes we made in the current revision of our manuscript.

Point 2. The introduction is based on 30 bibliography items and most of them are older than 4 years. I cannot agree that this section shows the current state of knowledge, especially in terms of speech recognition using machine learning approaches.

Response 2. First of all, thanks to this reviewer's comments, we reconsidered more carefully how our paper is positioned within the scope of signal processing and speech recognition agenda. In fact, the aspects of automated speech recognition (including machine learning approaches) are out of scope of our work. That's why, to avoid further misunderstanding, we revised the title to "Speech Processing for Language Learning: A Practical Approach to Computer-Assisted Pronunciation Teaching". However, we admit that the previous revision of our manuscript lacked clear positioning of our project in scope of current state-of-the-art research in the area. To fix this issue, We extended the material presented in the article, and significantly enhanced the article's bibliography. The extension includes the new section "Literature review" (p.4), subsections 3.2 "Prosodic similarity evaluation", and 4.3 "Prosodic similarity evaluation assessment".

Point 3. Section 2 shows the proposal, but there is no mathematical model, no novelty of the approaches [...] If it is an application proposal, the full background also should be presented. Especially, the novelty against existing solutions should be underlined here.

Response 3. We did the following steps in order to address this comment:
1) For better positioning of our work in frame of existing cross-disciplinary research on using speech processing approaches within the scope of creating systems for language learners (and to avoid understanding of our contribution in scope of speech recognition models), the Introduction was revised and a Literature review section was added.
2) We added the subsection 3.2. "Prosodic similarity evaluation" presenting the formal mathematical models used to evaluate language learner's pitches against the suggested model's pitches. These models include a set of CRQA metrics which were shown to display the effects of learner and model synchronization and approximation. These metrics were proven to be efficient for speech emotion recognition or conversation behaviour studies, but were never used in the domain of language learning.
3) Correspondingly, we wrote the subsection 4.3. "Prosodic similarity evaluation assessment", where we presented CRQA feature evaluation by two binary classifiers. Both models showed an accuracy increase when CRQA metrics RR and DD were used collectively with DTW scores to tell the difference between "good" and "poor" attempts. We observed the decrease in false positives and false negatives in comparison with pure DTW scoring by approximately 20%.

Point 4. The authors focused on the application interface, XML, etc.

Response 4. In response to the comment on application interfaces, let us kindly note, that, since the application interfaces have been described in our previous papers, the current one does not include much discussion about interface issues, except perhaps Figure 1(b), so we have been a little bit confused by this comment.

Point 5. The experimental section does not examine the proposal. There is no proper analysis of components, no comparison with other solutions from the last 3-4 years, and no deep discussion based on statistical analysis about results.

Response 5. The newly included Subsection 4.3. presents our analysis of experimental data collected at the assessment stage. As all issues mentioned in point 3, such as component analysis, related CAPT systems comparison and statistical evaluation, were analyzed and reported in our previous works, we focused only on specific questions, concerning the current updates. These include: (1) voice activity detection (probably, not described thoroughly because this part is based on related papers and seems rather straightforward); (2) CRQA approach which is presented and is new for language learning systems, the results of feature engineering and discriminating ability of DTW+CRQA are given in Section 4.3; (3) Transcription and rhythm reconstruction algorithms are evaluated through the comparison, numerically expressed as Levenstein distance, with reference corpora bearing the same information and labelled by human experts, which is a conventional assessment practice in the area of CAPT design.

Round 2

Reviewer 1 Report

no further comments

Author Response

Thank you for the positive opinion. We re-checked the text one more time to address the spelling and style issues as much as possible.

Reviewer 3 Report

The quality of the paper is much higher than before. I still have some issues which should be improved in this round of review:
1) The used bibliography is still in many parts outdated. Update the references to the current state of research, mainly from the last 3-4 years.
2) Add some analysis of time/computational complexities.
3) Compare your results with other state-of-art solutions.

Author Response

1) The used bibliography is still in many parts outdated. Update the references to the current state of research, mainly from the last 3-4 years.

As suggested, we have supplemented our references with several articles and books from 2016 onwards. To show the connection and support of the ideas expressed in earlier works, we would like to keep our original sources in the reference list. We acknowledge that recent references can supersede dated references, but we feel that some of our less recent references are seminal. References 34, 35, 39, 42,56, and 57 on the reference list are cases in point. These papers have lost none of their relevance and have greatly impacted the present research due to their excellent exposition of fundamental ideas either in pronunciation training, or non-linear dynamics, or speech technologies.

2) Add some analysis of time/computational complexities.

The discussion of methods (Section 3) mentions the complexity of core algorithms like voice activity detection (linear), syllable construction (linear), ASR decoding graph (linear), Levenstein distance (linear) and DTW/CRQA scorings which both need $O(N^2)$ computations in the worst case. All the other algorithms of the present design, e.g. pitch detection, interpolation, smoothing, etc. are of polynomial complexity. We have to pay attention to the complexity of algorithms we choose, because of system performance requirements for response time when working at mobile target devices. In some cases, it is necessary to sacrifice accuracy for performance in order to keep the whole design simple yet responsive.

3) Compare your results with other state-of-art solutions.

Because of a great variety of technical implementations of CAPT systems, pronunciation feature sets to focus on, target languages, and groups of users, the comparison in conventional sense is neither possible, feasible, or consistent.

Even partial comparison is problematic because CAPT performance has not been clearly defined so far and the authors working in the same direction do not have any agreed performance metrics. There is also a large variability in the evaluation approaches adopted. Pennington [29] contrasts the technological performance of modern CAPT systems with their didactic or pedagogical usability and highlights that the former does not imply the latter by the nature of the case.

Nonetheless, in response to the reviewer's suggestion, in Section 2, we made an effort to examine the architectures and functionality of the closest solutions. In one of our previous works[65] we evaluated StudyIntonation functionality using a set of criteria which are worked out in [64] exactly for the purpose of comprehensive assessment of CAPT systems. For a stricter peer-group comparison we should choose reference systems from those having the following specific features: it should be a free mobile CAPT system for English (1); it should offer both phonetic (segmental) and prosodic (suprasegmental) activities (2); it should provide visual feedback for the utterances in different speech situations (3); and it should implement the specific chain of algorithms (3). To the best of our knowledge, none of the other existing CAPT systems incorporates all these features. The solutions discussed in Section 2 have the maximum intersection of items mentioned above and are architecturally similar to our design. 

Round 3

Reviewer 3 Report

It can be accepted in the current form.